# Rotavirus A Inoculation and Oral Vitamin A Supplementation of Vitamin A Deficient Pregnant Sows Enhances Maternal Adaptive Immunity and Passive Protection of Piglets against Virulent Rotavirus A

**DOI:** 10.3390/v14112354

**Published:** 2022-10-26

**Authors:** Juliet Chepngeno, Joshua O. Amimo, Husheem Michael, Kwonil Jung, Sergei Raev, Marcia V. Lee, Debasu Damtie, Alfred O. Mainga, Anastasia N. Vlasova, Linda J. Saif

**Affiliations:** 1Center for Food Animal Health, Department of Animal Sciences, College of Food Agricultural and Environmental Sciences, The Ohio State University, Wooster, OH 44691, USA; 2Department of Veterinary Preventive Medicine, The College of Veterinary Medicine, The Ohio State University, Wooster, OH 44691, USA; 3Department of Animal Production, Faculty of Veterinary Medicine, University of Nairobi, Nairobi 00625, Kenya; 4Department of Immunology and Molecular Biology, School of Biomedical and Laboratory Sciences, College of Medicine and Health Sciences, University of Gondar, Gondar, Ethiopia; 5The Ohio State University Global One Health LLC, Eastern Africa Regional Office, Addis Ababa, Ethiopia; 6Department of Public Health, Pharmacology and Toxicology, Faculty of Veterinary Medicine, University of Nairobi, Nairobi 00625, Kenya

**Keywords:** Rotaviruses, lactogenic, immunity, maternal immunization, vitamin A, swine model

## Abstract

The aim of this study was to determine the impact of vitamin A deficiency (VAD)/supplementation (±VA) and group A RV (RVA) maternal immunization of RVA seropositive multiparous pregnant sows, on their immune responses (anamnestic response) and on passive protection of their piglets against RVA challenge. Our results showed that VAD- mock sows had increased RVA RNA shedding at 1–5 days post piglet RVA challenge, and their litters had increased RVA shedding and diarrhea frequency throughout the experiment. VAD decreased memory B cell frequencies while VA supplementation increased RVA specific IgA/IgG antibody (Ab) secreting cell (ASC) numbers in blood, milk, and tissues of RVA inoculated VAD sows. The increased numbers of RVA specific IgA/IgG ASCs in blood, milk/colostrum, intestinal contents, and tissues in VA supplemented VAD sows, suggest a role of VA in B cell immunity and trafficking to tissues. We also observed that RVA inoculated sows had the highest viral neutralizing Ab titers in serum and milk while VA supplementation of VAD sows and RVA inoculation increased IgA^+^ B cell frequencies in sow colostrum. In summary, we demonstrated that daily oral VA-supplementation (2nd trimester-throughout lactation) to RVA inoculated VAD sows improved the function of their gut-mammary-IgA immunological axis, reducing viral RNA shedding, diarrhea, and increasing weight gain in suckling piglets.

## 1. Introduction

Rotavirus (RV) is the main pathogen implicated in severe dehydrating diarrhea in children (<5 years) and young animals [1]. RVs infects the small intestines where they replicate in intestinal villi leading to functional and anatomical loss of these absorptive cells [2,3]. Yearly, ~200,000 children under 5 years of age, mainly in the middle- and low-income countries succumb to RVA diarrhea [4]. Due to the severity of RVA infection during the initial days of life, the major source of protection is through the passively acquired maternal antibodies (Abs) from colostrum and milk or in utero (humans). This protection is important especially during this sensitive time when the neonatal immune system is not fully mature to eliminate invading pathogens [5]. Moreover, the protective effects of passively acquired Abs in early life may persist into adulthood [6]. Therefore, maternal vaccination would protect the young against RVA infection until an age when their immune system responds effectively to ameliorate the disease severity. Maternal vaccination is valuable due to the fully mature immune system of the mother capable of mounting protective immune responses transferrable to neonates in utero and during lactation.

Previous studies have demonstrated that passive Abs inhibit infection and neutralize virus, minimizing virus replication and reducing shedding [7,8]. Therefore, the effects of maternally derived passive Abs (both serum Abs and milk-associated Abs in the intestine) should be considered when developing efficacious oral vaccines and immunization programs against RVAs for infants and young animals [9,10]. Clinical studies have reported reduced rates of postvaccination seroconversion in children with higher titers of maternal passive Abs in serum. Newborn piglets are agammaglobulinemic with inadequate cellular immunity until at least four weeks of age; hence they rely solely on maternal lactogenic passive immunity for responding to infectious agents early [11,12,13]. Therefore, boosting pre-existing maternal immunity against specific pathogens through exposure to the pathogen or immunization is a valuable approach for preventing disease in piglets reducing disease severity. Previously, maternal immunization has been shown to protect piglets from several infectious diseases including erysipelas, swine influenza, neonatal colibacillosis, Aujesky’s disease, atrophic rhinitis, necrotizing diarrhea, and Senecavirus infection [14,15]. Bohl, Saif and coworkers observed increased levels of TGEV neutralizing Abs and secretory IgA (sIgA) in milk of sows that were orally inoculated/naturally infected with TGEV than those that received TGEV through intramuscularly injection [16,17]. Similarly, piglets of the former sows showed increased TGEV-specific IgA Abs levels and higher survival rates [16,17,18]. These observations were the first description of the gut-mammary gland (MG)-sIgA axis that led to the concept of a common mucosal immune system [19,20,21]. Moreover, recently Yang and colleagues observed that vaccination of sows that received a high 1st dose of Senecavirus vaccines had maternal Abs in mammary secretions for longer time-periods improving passive immunity of piglets [15]. However, they demonstrated that Senecavirus-vaccinated piglets were able to overcome the interference by maternal Abs when the maternal Ab levels had declined (at ~60–90 days old). The main challenge to implementing and optimizing maternal immunization on-farm is the lack of techniques for evaluation and quantification of the passive immunity provided by the mothers. In addition, the amount of maternal Abs needed for effective protection of piglets against specific pathogens are undefined. Moreover, sow serum Ab levels prepartum did not correlate with Ab levels in mammary secretions [15]. Therefore, techniques to quantify maternal immune responses during gestation and to predict the amount of passive immunity needed to protect the piglets require further investigation.

Vitamin A (VA) is essential in numerous biological conditions, including maintenance and modulation of immunity, and maintenance of epithelial tissues and mucosal homeostasis [22]. Therefore, VA deficiency (VAD) causes numerous immune alterations including poor development of lymphoid structures, reduced adaptive and innate immune responses, and ineffective response to vaccination, leading to increased vulnerability to infectious diseases and even deaths [23]. VAD is associated with high mortality and morbidity in low resource settings and populations, especially in Southeast Asia and Sub-Saharan African countries where there is less access to VA sufficient diet. The effects of VAD on vaccines or infections, including the RVA, is not well defined, although it is well established that VAD enhances the incidence of respiratory and enteric infections by diminishing the protective efficacy of the sIgA Ab response [24]. For example, a previous study showed that VAD decreased the relative numbers of plasma cells [18] thereby reducing the influenza-specific IgA Ab; however, influenza-specific IgG Ab response in serum was not affected by VAD [25]. Homing of IgA Ab producing cells to the gut and secretion of secretory IgA Abs are important in conferring protection in the gut [21,26,27]. Surman and colleagues demonstrated that a single dose of VA (retinyl palmitate/retinol) administered intranasally during immunization against Sendai virus (SeV), was adequate to correct the impaired upper respiratory tract IgA responses in VAD mice [28]. Similarly, the same authors showed that oral VA supplementation rescued mucosal IgA Ab responses to influenza vaccine in VAD mice [29,30]. McGill and colleagues recently demonstrated impaired immune response to respiratory syncytial virus (RSV) infection and vaccine in neonatal calves, characterized by increased proinflammatory cytokines in the lungs, and severe disease in VAD calves [31]. Furthermore, acute RSV infection caused reduction in serum and liver retinol, making even VA sufficient calves vulnerable to VAD. Retinoic acid (RA), an active metabolite derived from VA, increases the expression of α4β7, a gut homing receptor in B cells [32]. Hammerschmidt and coworkers showed that RA is essential to induce trafficking of protective T and B cells from skin-draining inguinal lymph nodes to the gut after vaccinating mice via the subcutaneous route, thereby protecting VAD mice against cholera toxin–induced diarrhea [33]. Similarly, our lab demonstrated that VA supplementation in the second trimester increased B cells expressing α4β7 and CCR10 in the MG in gilts after oral inoculation with PEDV [34].

There is adequate evidence from clinical studies and animal model research that a high incidence of diarrhea correlates with increased prevalence of VAD [35], and that VA supplementation reduces diarrhea frequency in infants, through the maintenance of the intestinal epithelial integrity and regeneration of the damaged mucosal epithelial cells [36,37,38]. Using the germ-free piglet model, our lab demonstrated that prenatally acquired VAD impairs innate and adaptive immune responses to virulent human RVA infection and oral attenuated human RVA vaccines, where we observed severe diarrhea post-challenge, coinciding with decreased serum RVA specific IgA Abs and intestinal RVA-specific IgA Ab secreting cells (ASCs) in vaccinated VAD pigs [39,40]. Recently, Xing and coworkers demonstrated an association between VAD and severe *Mycoplasma pneumoniae* pneumonia in children [41]. Although, Long and colleagues observed prolonged norovirus shedding following VA supplementation of VAD mice, they observed decreased norovirus-associated diarrhea incidence in the same group of mice [42].

The aim of our study was to evaluate the effect of VA status and RVA inoculation (which mimics maternal immunization) of RVA seropositive pregnant sows on their adaptive immune responses and passive protection of their piglets against RVA challenge. Currently, there is a lack of an efficacious RVA vaccine in middle- and low-income countries; hence our results provide new options for improving vaccination strategies against RVA and other enteric pathogens. An optimally designed maternal vaccination strategy will enhance maternal immunity and lactogenic protection needed to prevent RVA-associated diarrhea or mortality in the newborn. Additionally, effective maternal vaccination would benefit swine producers by lowering the cost of raising swine and pork production.

## 2. Materials and Methods

### 2.1. Virus

Virulent porcine RVA OSU (G5P [7]) (passaged in gnotobiotic pigs) diluted in minimal essential medium (MEM; Life Technologies, Grand Island, N.Y.), was used to inoculate sows at gestation day (GD)~90 and challenge piglets at day post-partum (DPP)~5. Attenuated OSU PRVA propagated in MA104 cells (Origin = grivet monkey {chlorocebus aethiops} sold by Microbiology Associates now known as Bio Whittaker, Walkersville, MD, USA) was used for ELISPOT and virus neutralization (VN) assays.

### 2.2. Experimental/Sampling Strategy

RVA seropositive sows (~100% of commercial swine) were assigned to VAD (*n* = 14) or VAS (*n* = 6) diets at ~GD30. A subset of VAD sows was given daily oral retinyl palmitate (30,000IU; Sigma-Aldrich, St. Louis, MO, USA) (VAD + VA, *n* = 7) starting at ~GD76 throughout the rest of gestation and lactation (Figure 1). RVA OSU [G5[P7]] strain was used to inoculate sows GD~90 (1 × 10^9^ FFU/mL) and piglets (1 × 10^8^ FFU/mL) at ~5 days postpartum (DPP~5). The control sow group was inoculated with sterile Minimum Essential Media (MEM).

### 2.3. Rotavirus Fecal Shedding (Sows and Piglets), Piglet Diarrhea Severity and Frequency and Weight Changes

Clinical signs of RVA infection in sows were recorded every other day following RVA inoculation and rectal swabs samples were collected for examining RVA RNA shedding by RT-qPCR using one step RT-qPCR Qiagen kit (QIAGEN, Germantown, MA, USA). Piglet weights were recorded at birth, post challenge day (PCD)0, PCD3, PCD7 and PCD12. Rectal swabs were collected from each piglet from PCD0–12 for RVA RNA shedding enumeration and fecal scores were recorded as follows: 0, normal; 1, pasty; 2, semiliquid; 3, liquid.

### 2.4. Isolation of MNCs

Mononuclear cells (MNCs) from ileum, mesenteric lymph nodes (MLN), spleen, blood and MG were isolated as described previously [34,43].

### 2.5. Isolation of MNCs from MG Secretions

The milk/colostrum samples were diluted 1:3 in sterile Dulbecco’s phosphate-buffered saline [DPBS, (pH 7.2, without calcium and magnesium), Life Technologies, Grand Island, NY, USA] to improve MNC recovery efficiency by dispersion of the fat micelles. Furthermore, before dilution, a separate portion of whole milk/colostrum (~50 mL) was used to obtain skim milk for subsequent whey processing. The diluted and undiluted milk/colostrum was centrifuged at 800× *g* at RT for 10 min for separation of cell pellet, skim milk and fat. The skim milk from the undiluted tube was retained and used to obtain whey as described below. The cell pellets were combined and washed once with DPBS and then re-suspended in 1 × Hanks’ Balanced Salt Solution (HBSS; Life Technologies, Grand Island, NY, USA), centrifuged 200× *g*, at RT for 10 min. The pellet was then resuspended in Percoll (Cytiva, Sweden) (43%), depending on pellet size, then split into 10 mL each in 15 mL tubes. Each tube was underlaid with 5 mL of 70% Percoll, then centrifuged at 1000× *g* at RT for 30 min without brake. The MNCs were collected at the interface of 43% and 70% Percoll and resuspended with 1 × HBSS then centrifuge for 10 min at 200× *g* and 4 °C. The resultant pellet was resuspended in 1–5 ml enriched RPMI (Life Technologies, Grand Island, NY, USA) depending on the size of the pellet, and the MNCs quantity and viability was measured in Cellometer by trypan blue (Thermo Fisher Scientific, Waltham, MA, USA) exclusion.

### 2.6. Whey Isolation

Skim milk was obtained as described above and centrifuged at 90,000× *g* for 60 min at 4 °C. Whey was collected from the middle layer and stored at −20 °C until tested. Whey was used to quantitate the IgA and IgG Abs in milk as described below.

### 2.7. Virus Neutralization Antibody Assay

A plaque reduction VN antibody assay was performed on serum and milk samples as described previously [7], with porcine RVA OSU. VN antibody titers were expressed as the reciprocal of the serum/milk dilution which reduced the number of plaques by >80%.

### 2.8. ELISPOT Assay for Virus-Specific ASCs

To quantify RVA-specific Ab secreting cells (ASCs) of different isotypes, ELISPOT assays were performed as described previously [43,44]. Briefly, acetone fixed OSU RVA-infected MA104 cells in 96-well plates (Nunc-Immuno, Nalge-Nunc, Rochester, N.Y.) were prepared. MNCs from each tissue sample were added to duplicate wells (5 × 10^5^ cells/well). Plates were incubated for 12 h at 37 °C in 5% CO_2_, then washed and biotinylated mouse mAb to pig IgG or pig IgA (Southern Biotech, Birmingham, AL, USA) was added to each well, and plates were incubated for 2 h at room temperature. Plates were washed, and Horseradish Peroxidase-conjugated streptavidin (Kirkegaard & Perry Laboratories Inc. [KPL], Gaithersburg, MD, USA) was added, followed by 1 h incubation at room temperature. The plates were washed, and spots developed with 3,3′,5,5′-Tetramethylbenzidine (TMB) substrate (KPL, Gaithersburg, MD, USA). The numbers of RVA virus-specific ASCs (blue spots) were determined in each well and the average numbers of spots were expressed relative to 5 × 10^5^ MNCs.

### 2.9. Rotavirus A-Specific Antibody (Ab) ELISA Assay

The lysate of cell culture adapted OSU G5P [7] RVA strain or mock-inoculated MA104 cells was prepared as a virus or control antigen for whole viral Ab ELISA, and the RVA specific IgG and IgA Ab ELISA was performed as described previously [39,40]. The titer for RVA-specific IgG and IgA Abs was expressed as the reciprocal of the highest dilution that had a corrected A_450_ value greater than the cut-off value (mean + three standard deviations of negative control samples).

### 2.10. Flow Cytometry

To assess the adaptive immune response in the sows, frequencies of B lymphocyte subpopulations were evaluated using a previously established protocol [45]. Briefly, to identify selected B cell subsets, isolated MNCs were stained with anti-porcine CD21-PE (clone BB6-11C9.6, Southern Biotech, Birmingham, AL, USA), anti-porcine CD2 (clone MSA4, King Fisher Biotech, St Paul, MN, USA), and anti-mouse (porcine cross-reactive) CD79β-FITC (clone AT1072, Bio-Rad, Hercules, CA, USA) mAbs. Additionally, mouse anti-pig IgA mAb (K61 1B4, Bio-Rad, Hercules, CA, USA) was used to evaluate the frequencies of IgA producing B cells among the MNCs. Finally, human (porcine cross-reactive) integrin alpha 4 beta 7(α4β7)/LPAM-1 (Clone # Hu117, R&D Systems, Minneapolis, MN, USA) and anti-mouse CCR10 (clone 248918, R&D Systems, Minneapolis, MN, USA) were used to examine the expression of gut homing markers (α4β7/CCR10) among the isolated MNCs. Suitable isotype control and secondary Abs were included in each assay. Stained samples were analyzed using Accuri C6 flow cytometer (BD Biosciences, San Jose, CA, USA) with acquisition set at 50,000 events. The frequencies of different cell populations were determined using CFlow software (AccuriC6 cytometers; BD Biosciences, San Jose, CA, USA).

### 2.11. Statistical Analysis

Statistical analysis was done using GraphPad Prism 8 (GraphPad Software, Inc., San Diego, CA, USA). Normalized weight change, fecal RVA RNA shedding post-challenge, and log transformed RVA-specific Ab titers (IgA and IgG) were compared using one-way analysis of variance (ANOVA). Frequencies of B cell subpopulations, VN titers between and within groups were compared using two-way ANOVA and Kruskal–Wallis rank sum test (*p* ≤ 0.05 was considered significant).

## 3. Results

### 3.1. Serum and Hepatic Vitamin A Levels

Sows fed VA deficient diet (VAD) had decreased hepatic VA levels compared with VAS and VAD + VA (supplemented VA start at GD~76) sows (Figure 2a), whereas VAD + VA and VAS sows had comparable hepatic VA levels. Expectedly, the analysis of serum VA levels among the sow groups revealed comparable levels among all three groups at GD~30, GD~76, GD~109 and DPP~21 (Figure 2b). However, the serum VA level was slightly higher in VAS vs. VAD + VA and VAD sows at all sampling time-points. Interestingly, in all the sow groups, the serum VA levels increased during the mid-gestation, decreased just prior to parturition (GD~109) before rising again after parturition (DPP~21). This could be due to late pregnancy stress or changes in pregnancy related hormones during this period.

Finally, we observed significantly higher (*p* < 0.001) hepatic VA levels of piglets born to VAS sows compared to VAD (±VA) sows. Additionally, the litters of VAD + VA sows had significantly (*p* < 0.01) higher hepatic VA levels than the litters of VAD sows (Figure 2c).

### 3.2. Rotavirus A Inoculation and VA Supplementation to VAD Sows during Pregnancy and Lactation Reduced Viral Shedding in Piglets

We observed significant differences (*p* < 0.05) in mean body weight gain between the litters of different sow groups following RVA challenge (Figure 3a). Although piglets of all the sow groups gained weight after RVA challenge, the weight gain of the piglets of mock VAD-Mock sows was lower throughout the experiment. RVA challenge of suckling piglets resulted in higher diarrhea frequency and severity (as reflected by mean cumulative fecal score) lowest mean body weight gain and increased peak viral RNA shedding in these piglets (Figure 3a,b, Table 1). There was significantly (*p* < 0.05) higher RVA RNA shedding in feces of the VAD-mock litters at PCD2–7 compared with the VAS + RVA and VAD + VA + RVA litters (Figure 3b).

Overall, VAS + RVA, VAD + RVA and VAD + VA + RVA litters had lower RVA RNA shedding throughout the experiment, coinciding with low diarrhea severity and frequency, and improved weight gain. Since the sows used in this study were RVA seropositive, none of the sows showed any RVA clinical signs post RVA inoculation in 3rd trimester, although only RVA inoculated sows shed post inoculation. However, during lactation, all mock sows had increased RVA RNA shedding after RVA challenge of their piglets (DPP~5/PCD0) compared with the sows inoculated with RVA during gestation (Figure 3c).

### 3.3. Vitamin A Supplementation Increased RVA Specific IgA/IgG ASCs in Blood, Milk, and Tissues of RVA Inoculated VAD (VAD + VA + RVA) Sows While Vitamin A Deficiency Increased Frequencies of Total Ig-Secreting B Cells in Milk and Various Tissues, but Decreases Numbers of Total Memory B Cells in Milk

To understand the impact of VA status and RVA inoculation in the 3rd trimester of multiparous RVA seropositive pregnant sows on immune responses and passively acquired protection of their piglets, we evaluated several adaptive immune responses in the blood, colostrum/milk, intestinal contents and selected tissues in the sows, as well as in blood and intestinal contents of the piglets. There were no appreciable differences in the RVA-specific IgA/IgG ASC numbers between VAD and VAS sows except for VAD + RV sows having consistently and often statistically the lowest numbers in blood, milk, and different tissues (Figure 4). Furthermore, VA supplementation of VAD + RV sows increased RVA-specific IgA/IgG ASC numbers in their blood, milk, and various tissues to the levels comparable to those observed in VAS sows or higher (Figure 4). Similarly, VA supplementation of VAD-mock sows increased RVA-specific IgA/IgG ASC numbers in their blood, milk, and various tissues; however, the effect was less pronounced and consistent than in VAD + RV sows (Figure 4). Of interest, the effect of sow RVA inoculation at GD~90 on RVA-specific IgA/IgG ASC numbers in blood, milk and various tissues was marginal and inconsistent.

The mean numbers of RVA specific IgA and IgG ASCs in blood were numerically or significantly higher in VAD + VA + RVA sows compared to the other groups prior to parturition and during lactation (Figure 4a,d). In comparison, the non-VA supplemented VAD + RVA sows had numerically or significantly (*p* < 0.05–0.001) lower mean numbers of RVA-specific IgA ASCs in blood prior to parturition and during lactation, respectively than in other groups (Figure 4a). We observed increased numbers of RVA-specific IgA and IgG ASCs in milk of the VA supplemented VAD + VA (±RVA) sows throughout lactation, except at DPP~5 where the RVA specific IgA ASC were comparable among VAS-mock, VAD + VA + RV and VAD-mock groups (Figure 4b,e). VAD + VA + RVA sow groups had increased RVA specific IgG ASC in milk prior to piglet RVA challenge at DPP~5/PCD0 (Figure 4e). Overall, throughout lactation (except at DPP21/PCD14), VAD + RVA and VAS + RVA sows generally had the lowest RVA specific IgA ASC numbers in milk. Examination of selected tissues (spleen, MG, MLN and ileum) demonstrated that RVA specific IgA ASCs were significantly (*p* < 0.05–0.01) elevated in all the tissues of VAD + VA (±RVA) sows, with highest numbers observed in ileum that reflected the higher magnitude, but generally similar profiles to those in milk (Figure 4c). A similar trend was observed for RVA specific IgG ASCs (Figure 4f). On the other hand, VAD + RVA and VAS + RVA sows had the lowest IgA ASC (+VAD-mock for IgA ASC) and IgG ASC numbers in all tissues. In general, our data showed that both RVA specific IgA and IgG ASCs were elevated in blood, milk and tissues of VAD + VA (±RVA) sows compared with VAD + RVA sows, suggesting a role of VA in modulating these ASC responses.

On the other hand, VA supplementation increased total numbers of Ig-secreting B cells in milk and ileum of VAD + RV sows (VAD + RV vs. VAD + VA + RV) (Appendix A). VAD-mock sows generally maintained higher frequencies of total Ig-secreting B cells in milk throughout lactation (Appendix A). The ratios of activated to naïve B cells were highest in milk of VAS-mock sows compared with all other groups. However, this ratio was the highest in ileum and MG of VAS + RV, VAD + VA-Mock and VAD + RV sows, and MLNs and spleen of VAD + VA-Mock sows. In addition, the ratios of activated to naïve B cells were significantly higher in VAD + RVA sows than in VAD-mock, VAS-mock and VAD + VA + RVA sows in MG.VAD status and the lack of RVA inoculation of VAD-Mock and VAD + VA-Mock resulted in the lowest numbers of memory B cells in sow milk. Finally, the highest numbers of memory B cells in the MG were associated with the RVA inoculation of sows (VAD + RV and VAS + RV) (Appendix A).

### 3.4. Rotavirus A Specific IgA Antibody Titers Were Increased in Serum in All Sows Prior to Parturition and in Most Sows following Piglet Challenge

Pre-parturition, no consistent differences were observed in the levels of RVA-specific serum IgA Ab titers among the sows. VAS + RVA sows had the highest RVA-specific serum IgA Ab titers (Figure 5a) after parturition confirming a role of the RVA sow booster inoculation and adequate VA levels in IgA Ab production. In contrast, no consistent trends were observed for the mean RVA-specific IgG Ab titers in serum of different sow groups pre- or post-parturition (Figure 5d). Additionally, our results showed that RVA-specific IgA Ab titers were increased in serum of all the sows, irrespective of VA or RVA inoculation, immediately prior to parturition (GD~109) (Figure 5a). However, after parturition (DPP~5), we observed a significant increase in mean RVA-specific IgA Ab titers in RVA inoculated sow groups compared to mock groups (*p* < 0.05).

Evaluation of the RVA Ab titers in milk did not demonstrate any significant differences or clear trends in RVA-specific IgA Ab titers among the groups (Figure 5b); however, there were significant differences (*p* < 0.05–0.001) in mean RVA specific IgG Ab titers among the groups, with VAD + VA + RVA, VAS + RVA and VAS-mock groups showing marginally increased RVA specific IgG Ab titers in colostrum. Moreover, RVA-specific IgG Ab titers were significantly lower in milk of VAD + RVA and VAD + VA-Mock groups (Figure 5e).

Examination of the small and large intestinal contents (SIC, LIC) collected at DPP21/PCD14 demonstrated that there were no significant differences between groups, but the RVA-specific IgA Ab titers were increased in LIC of VAS + RVA, VAD + VA + RVA and VAS-mock sow groups compared with the VAD (±RVA) groups with the lowest in the VAD-mock group (Figure 5c). On the other hand, RVA specific IgG Ab titers were elevated in SIC of VAD + VA + RVA and mock (VAS/VAD) sows compared with VAS + RVA and VAD + RVA sows (Figure 5f). The IgG Ab titers were highest in LIC of VAD + VA + RVA and VAD mock sows compared with the other groups (Figure 5f).

### 3.5. Adequate Maternal Vitamin A Status and RVA Immunization during Gestation Increases RVA-Specific IgA and IgG Ab Titers in Piglets

Our data demonstrated that adequate maternal VA status and RVA immunization (VAS + RVA sows) resulted in the consistently highest serum and intestinal (SIC and LIC) RVA-specific IgA and IgG Ab titers (except LIC), while VAD-Mock sows generally had the lowest Ab titers (Figure 6). However, the effect of VA supplementation was inconsistent throughout the lactation (evident for serum samples) and most pronounced in piglets born to mock inoculated sows (VAD-Mock vs. VAD + VA-Mock) (Figure 6).

Further, there were significant differences (*p* < 0.05–0.001) in RVA-specific IgA/IgG Ab titers in serum of piglets among the groups before challenge (PCD 0) (Figure 6a,b) with piglets of RVA inoculated sows having increased Ab levels compared with piglets of mock inoculated sows. Piglets of RVA inoculated sows maintained high levels of RVA-specific IgA Ab titers throughout lactation (Figure 6a), an indication that maternal inoculation enhanced the passively acquired Abs. However, RVA-specific IgA Ab titers were significantly (*p* < 0.05–0.001) increased in SIC and LIC of piglets from VAS + RVA and VAD + RVA sows compared with the litters of other groups (Figure 6c), whereas RVA specific IgG Ab titers of these 2 sow groups were significantly lower (*p* < 0.01) in SIC (Figure 6d).

### 3.6. Sow RVA Inoculation, but Not Vitamin A Supplementation was Associated with The Highest Titers of RVA-Specific VN Abs

Surprisingly, VAD or VA supplementation did not have substantial effects on the levels of RVA-specific VN Ab titers in sow serum and milk or piglet serum (Figure 7). In contrast, sow RVA inoculation was associated with the highest RVA VN titers (VAD + RV and VAS + RVA sows). Our data demonstrated that VAD + VA-mock sows and their piglets had the lowest RVA VN Ab titers in serum throughout the experiment (Figure 7a,c). Overall, all RVA-inoculated sows had numerically (VAD + RV and VAD + VA + RV) or significantly (VAS + RVA) elevated RVA VN Ab titers in colostrum compared to non-inoculated sows (Figure 7b), suggesting that maternal inoculation with RVA during late pregnancy boosted the passively acquired maternal Abs. These Abs decreased immediately after parturition (DPP~5); however, they again increased after the piglets were challenged with RVA (PCD7 and 14) in the VAS + RVA and VAD + RVA, but not in the VAD + VA + RVA sows (Figure 7b), probably due to RVA reinfection of the sows in the former groups. Evaluation of RVA VN Abs in piglets demonstrated that the litters of VAD + RVA and VAS + RVA sows had elevated VN Ab titers prior to RVA challenge (DPP3–5/PCD0), and the VAD + VA-Mock piglets had the lowest VN Ab titers (Figure 7c). The VN Ab titers decreased in all the litters by the end of experiment.

### 3.7. VA Supplementation and RVA Inoculation Synergistically Increased the Frequency of IgA^+^ B Cells in Colostrum; However, VAD-Mock Sows Maintained the Highest Numbers of Milk IgA^+^ B Cells throughout Lactation

Although very low compared with milk, VAD + RV or VAD + VA-mock sows had significantly (*p* < 0.01) elevated frequencies of IgA^+^ B cells in blood during gestation compared with the other groups; however, although the levels decreased after parturition, they still maintained higher levels compared to most other groups during lactation (Figure 8a). Overall, analysis of the frequencies of IgA^+^ B cells in milk demonstrated that they were very high in all sow groups, with the VAD + VA + RVA group having the highest frequencies in colostrum (DPP0), while VAD-mock sows had the highest frequencies in milk throughout lactation (Figure 8b). Furthermore, the frequencies of IgA^+^ B cells in colostrum (DPP0) were significantly higher in VAD-Mock and VAD + VA-Mock sow groups. At euthanasia (DPP~21/PCD14), IgA^+^ B cell frequencies were highest in the ileum of VAS + RV and VAD + RV sows compared with the other groups (Figure 8c). Surprisingly, MG and spleen had the lowest, but comparable IgA^+^ B cell frequencies compared with the other tissues in all the sow groups (Figure 8c).

### 3.8. VA Sufficient Status and RVA Inoculation (VAS + RVA) Significantly Increased the Frequencies of Gut Homing (CCR10+) B Cells in Blood of Sows

The frequencies of α4β7^+^ cells were elevated in blood throughout the experiment, except the VAD-mock group where the frequencies were lowest at GD~109 and DPP~5 (Figure 9a). However, the frequencies of α4β7^+^ cells in milk were the highest in the VAD-mock group while the VAS-mock and VAD + VA-mock groups showed extremely low frequencies of these cells from DPP~5 throughout lactation (Figure 9b). Although no statical differences were observed, there were increased frequencies of α4β7^+^ cells in MLN and spleen (primary lymph nodes) of all sow groups (Figure 9c). Interestingly, the frequencies of α4β7^+^ cells were extremely low in the ileum of all the sows compared with other tissues (Figure 9c).

Conversely, CCR10^+^ cell frequencies were significantly higher (*p* < 0.001) in blood of the VAS + RVA sow group throughout gestation and lactation, while the other groups had comparable low frequencies of CCR10+ cells in blood throughout the experiment (Figure 9d). VAD + VA + RVA and VAD-mock sows maintained higher or significantly higher (*p* < 0.05) frequencies of CCR10^+^ cells in milk throughout lactation (Figure 9e), whereas VAS + RVA and VAD + RVA sows had the lowest frequencies. The VAS-mock sows had lower frequencies of CCR10^+^ cells in the beginning of lactation (DPP0, 3–5); however, the levels increased steadily and became the highest at DPP~12/~21 (Figure 9e). Although no statical differences were observed, VAS-mock, VAD-mock (±VA), VAD + RVA and VAD + VA + RVA sows had higher and comparable frequencies of CCR10+ cells in all tissues at euthanasia (Figure 9f), whereas the VAS + RVA group had the lowest frequencies of these cells in all the tissues examined.

## 4. Discussion

Understanding the mechanisms of the immune response to virus infection or vaccination is important to evaluate the vulnerability of individuals or human and animal populations and to comprehend the correlation between infection/vaccination, the host immune response, clinical signs, and protection against pathogens. Abs play an important role in the host defense as demonstrated by the (lactogenic) protection provided to neonates by maternal Abs [21,46]. In this study, we assessed the anamnestic immune responses in RVA seropositive multiparous pregnant sows that were inoculated with porcine RVA in the 3^rd^ trimester and evaluated the passive lactogenic protection of their nursing piglets against RVA challenge. Because of the lack of transplacental transfer or maternal Abs in sows, maternal Abs in colostrum/milk are essential for lactogenic protection of neonatal piglets until endogenous Abs can be generated in adequate quantities [47]. Animal research has provided substantial evidence that optimal VA levels are needed for efficient Ab responses to many pathogens and that VAD during infection negatively affects immune responses [48]. Thus, we further evaluated whether VA supplementation to VAD pregnant sows (RVA seropositive +/− RVA booster inoculation) could enhance their B cell immune responses and passive protection of their litters.

We have established VAD as previously by feeding a group of sows a VAD diet starting at GD~30 until the end of experiment (DPP~21) [45]. The slight decrease in serum VA levels in the 3^rd^ trimester prior to parturition (GD~109) observed in the current study in all sow groups before increasing after parturition (DPP~21), suggests that there is an increased requirement of VA during the 3rd trimester possibly due to changes related to pregnancy including changes in concentration of pregnancy associated hormones [49], while significantly lower hepatic VA levels in piglets in VAD litters shows that maternal VA status affected piglets in utero and during lactation.

IgA Ab is an essential functional effector of humoral adaptive immunity at mucosal sites [50], and the amounts of RVA-specific IgA Abs in pigs correlate with the protection against RVA [43,51,52]. Previous studies have shown that severe and prolonged diarrhea and increased human RVA fecal shedding titers were observed post-RVA challenge in VAD pigs that coincided with reduced RVA specific IgA/IgG Ab titers in serum/intestinal contents and reduced RVA specific IgA ASCs in intestinal tissues pre-/post-RVA challenge [39,40]. In this study, RVA inoculation and VA supplementation to VAD sows during the 3rd trimester led to reduced diarrhea frequency and viral RNA shedding titers in piglets indicating a combined effect of amnestic response to RVA and/or VA supplementation [53,54,55], coinciding with increased levels of RVA-specific IgA ASCs, IgA and IgG Ab titers and VN Abs in mammary secretions (milk/colostrum). These observations are consistent with our hypothesis that RVA IgA ASCs, VN Abs and Abs titers in milk/colostrum are crucial for lactogenic immune protection in neonatal piglets against RVA infection. Since sows were RVA seropositive, they developed no clinical RVA disease; however, mock sows had increased RVA RNA shedding during lactation compared with RVA inoculated sows. This suggests that the mock sows became infected after their offspring were challenged with RVA. Moreover, the increase in the mean numbers of RVA specific IgA/IgG ASCs in blood, milk/colostrum, intestinal contents, and tissues (spleen, MG, MLN, ileum) in VA supplemented VAD sows, indicate a role of VA in B cell immunity and trafficking of cells to tissues [48]. Similarly, RVA specific IgA/IgG and VN Ab titers were increased in serum/colostrum of RVA inoculated sows but not in mock sows, indicating an anamnestic response to RVA, thus, increasing RVA specific Ab production, and immune cell trafficking/accumulation in the MG.

RVA specific IgA/IgG Ab were increased in the VAS + RVA and VAD + VA + RVA groups in serum, intestinal contents, and colostrum indicating that RVA inoculation (maternal immunization) enhances the production of maternal Abs and passive lactogenic transfer of these Abs to offspring as observed by their increased weight gain, reduced viral RNA shedding titers and decreased severity and frequency of diarrhea. The current results are consistent with our previous observations in Gn pigs where intestinal IgA ASC and IgA Abs were involved in protection against RVA infection [19,39,56,57], and that VA and RVA inoculation impact the development of systemic B cell and innate immune responses [53,54]. Moreover, these responses coincided with increased RVA-specific IgA/IgG Ab responses in serum and intestinal contents. Additionally, VAD + VA + RVA and VAD + VA-mock sows had elevated frequencies of Ig-secreting B cells in milk throughout lactation or only in early lactation, respectively, indicating the role of VA in the production of Ig secreting B cells that were reduced in non-supplemented VAD-mock sows. The increase in the ratios of activated to naive B cells in MG and ileum of RVA inoculated sows suggest the importance of an anamnestic immune response in trafficking of these cells to these tissues.

VAD-mock sows had reduced frequencies of memory B cells in milk throughout lactation, while VA supplementation partially rescued this effect, indicating an essential role of VA in memory B cell responses. Both VAS-mock and VAD + VA-mock sows had elevated frequencies of activated vs. naïve B cells throughout lactation, suggesting that adequate VA is needed to generate immune responses, as noted by increased RVA specific IgA/IgG Ab in the circulation in these groups. Besides, VAD + VA-mock sows had elevated frequencies of memory B cells in spleen/MLN and IgA+ B cells in colostrum, suggesting that VA regulates generation of these cells in systemic and mucosal immune sites [58,59].

Frequencies of IgA+ B cells in VAD + VA + RVA sows were elevated in colostrum while VAS + RVA and VAD + RVA sow groups had elevated IgA+ B cells in ileal MNCs at DPP21/PCD14, indicating that the RVA anamnestic response is more pronounced in ileum and colostrum. VAD-mock sows had elevated frequencies of IgA+ B cells in milk, that suggests a compensatory or counter-regulatory effect of VAD, where these sows mobilize hepatic VA reserves to maintain normal serum VA concentrations [60] or an indirect negative effects of gut microbiota mediated by VAD [61].

Considering previously published data [62], we hypothesized that VAD may impact the homing of B lymphocytes. For instance, the expression of α4β7^+^ integrin on B cells is dependent on VA [62] and is responsible for cellular migration in the intestine and mucosa lymphoid tissues by interacting with its receptor MAdCAM-1 [63]. VAD-mock sows had reduced, while VAS + RV sows had elevated frequencies of α4β7^+^ cells in blood, indicating that adequate VA levels are required to recruit gut homing cells into systemic circulation [62]. Interestingly, VAS-mock and VAD + VA-mock groups sows had reduced frequencies of α4β7^+^ cells in milk during lactation, suggesting the importance of the anamnestic RVA response for efficient recruitment of the gut-homing cells. Surprisingly, elevated frequencies of α4β7^+^ cells were observed in MLN, MG and spleen while reduced in ileum among all the groups. This suggests a reduction in ileum due to their homing to other mucosal tissues, which requires further investigation.

Taken together our results are consistent with earlier research demonstrating the effects of VAD on the immune response to infections in various hosts. For example, VAD significantly increases the mortality rate as a result of measles infection [64] and/or diarrhea [65] while VA supplementation decreases the morbidity from these and other infectious diseases, suggesting that VAD has a crucial impact on immunity. Furthermore, previously it was demonstrated that VAD reduced Ab titers against tetanus toxin and impaired Ab responses were rescued in children and a mouse model by VA supplementation [66,67]. VAD has also been shown to reduce antigen-specific IgG responses in mouse models [68,69]. Similarly, VAD reduced the levels of Abs against pneumococcal polysaccharide antigen in a murine model [70], suggesting that there is a correlation between VA and the production of virus-specific Abs. Therefore, this is the first study to our knowledge to show that maternal VAD can result in piglet VAD and decreased protective B cell and Ab responses and lactogenic passive immunity to RVA in a conventional pig model.

In conclusion, we showed that sows that received RVA inoculation (maternal immunization) during pregnancy delivered adequate lactogenic immunity to their piglets with increased passive protection, and that VA supplementation of VAD + RVA sows (VAD + VA + RVA) during gestation and lactation enhanced sow immune responses (anamnestic responses) and passive protection of their offspring. Our findings are important for understanding the immunomodulatory role of VA during infection/vaccination and the impact of maternal VAD and immunization (booster) on neonatal passive immune protection. Furthermore, our novel data provide critical insights for development of effective immunization programs such as human and swine RVA vaccination programs that may lead to improved control over RVA infection neonates.

## Figures and Tables

**Figure 1 viruses-14-02354-f001:**
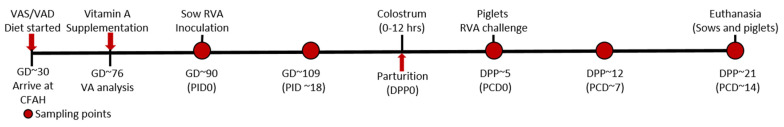
Experimental design to determine the impact of Vitamin A status and RVA inoculation of RVA seropositive pregnant sows on their immune responses and passive protection of their piglets (GD—Gestation Day, PID—Post Inoculation Day, PCD—Post Challenge Day, VAS—Vitamin A sufficient diet, VAD—Vitamin A deficient diet, RVA—rotavirus A).

**Figure 2 viruses-14-02354-f002:**
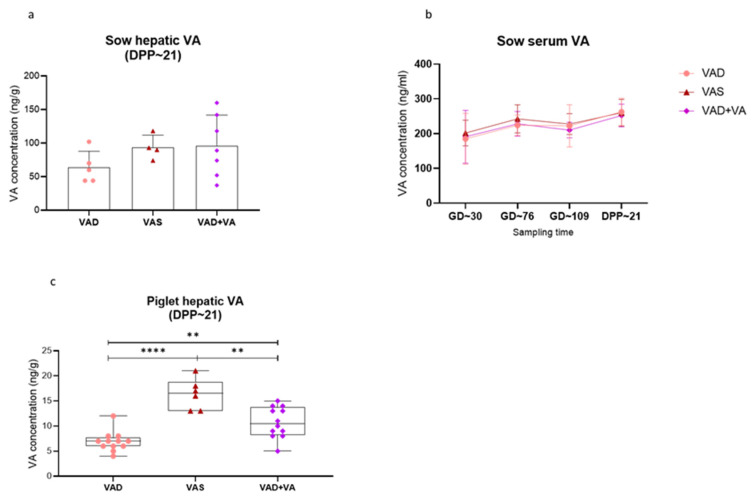
Pregnant sows were placed on different VA regimen diets (VAD, VAS and VAD + VA). Blood was collected for serum processing and livers at euthanasia (Day Post-Partum ~21). VA levels were determined by HPCL at Iowa State University Veterinary Diagnostic Lab (Ames, Iowa). (**a**) Mean hepatic VA concentrations in sows. (**b**) Mean serum VA concentrations in sows. (**c**) Mean hepatic VA concentrations in piglets (** *p* < 0.01; **** *p* < 0.0001).

**Figure 3 viruses-14-02354-f003:**
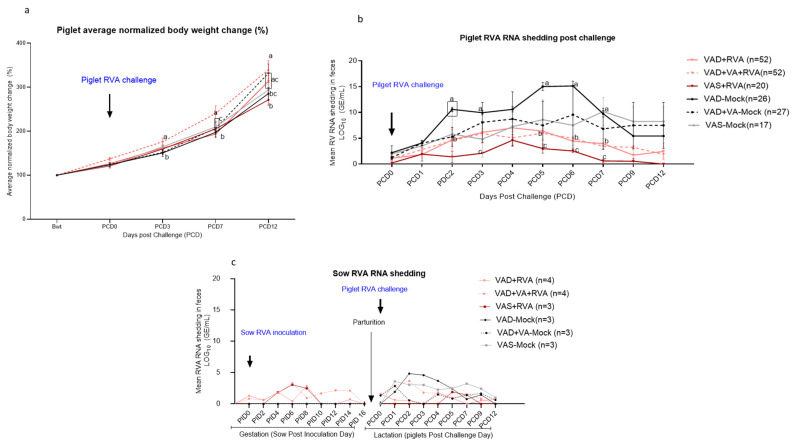
Average normalized piglet body weight change from birth to RVA Post Challenge Day (PCD) ~12 (**a**), RVA RNA shedding was determined by RT-qPCR and expressed as log_10_ GE/mL. Viral RNA shedding in piglets was measured at PCD0–12 (**b**), and sows RV RNA shedding post-sow inoculation and post-piglet challenge (**c**). *a*, *b*, and *c* represent significant differences among the groups (*p* ≤ 0.05).

**Figure 4 viruses-14-02354-f004:**
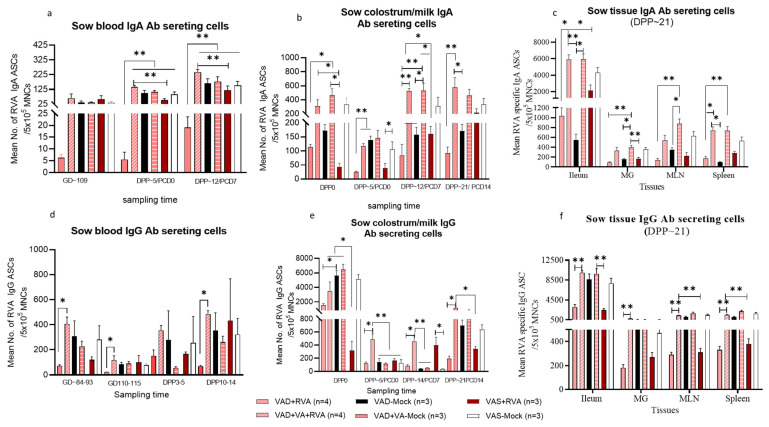
Mean RVA IgA antibody secreting cells (**a**–**c**) and mean RVA IgG antibody secreting cells (**d**–**f**) in Blood, milk, and tissues MNCs of RVA/mock inoculated sows fed VAS, VAD (±VA) diets during gestation and lactation. DPP—days postpartum, GD—gestation day, PCD—post challenge day (* *p* < 0.05; ** *p* < 0.01). MLN—mesenteric lymph node, MG—mammary gland.

**Figure 5 viruses-14-02354-f005:**
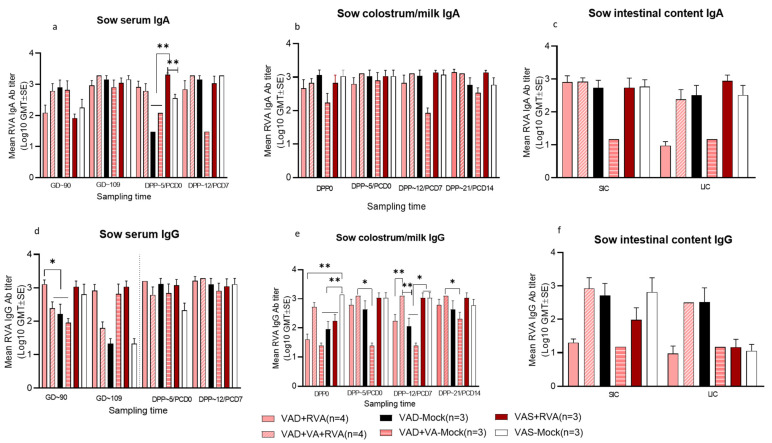
Mean RVA specific IgA antibody titers (**a**–**c**) and mean RVA IgG antibody titers (**d**–**f**) in serum, milk and intestinal contents (SIC/LIC) of RVA/mock inoculated sows fed VAS, VAD (±VA) diets during gestation and lactation (* *p* < 0.05; ** *p* < 0.01). SIC—small intestinal contents, LIC—large intestinal contents.

**Figure 6 viruses-14-02354-f006:**
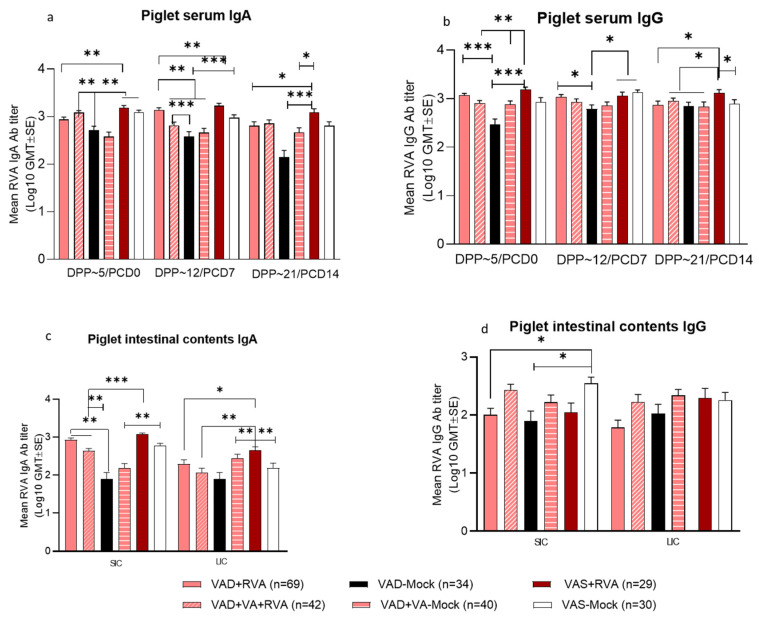
Mean RVA specific IgA antibody titers (**a**,**c**) and IgG antibody titers (**b**,**d**) in serum (**upper**) and intestinal contents (**lower**) of piglets of RVA(OSU)/mock inoculated sows fed VAS and VAD (±VA) diets during gestation and lactation (* *p* < 0.05; ** *p* < 0.01; *** *p* < 0.001), SIC—small intestinal contents, LIC—large intestinal contents.

**Figure 7 viruses-14-02354-f007:**
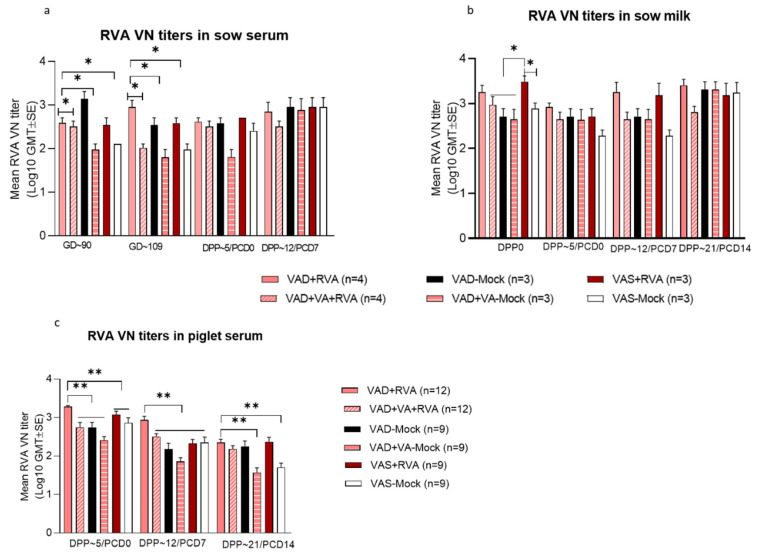
Mean RVA virus neutralizing (VN) antibody titers in sow serum (**a**), milk (**b**) and piglet serum (**c**) of mock and RVA(OSU) inoculated sows fed VAS and VAD (±VA) diets during gestation and lactation (* *p* < 0.05; ** *p* < 0.01).

**Figure 8 viruses-14-02354-f008:**
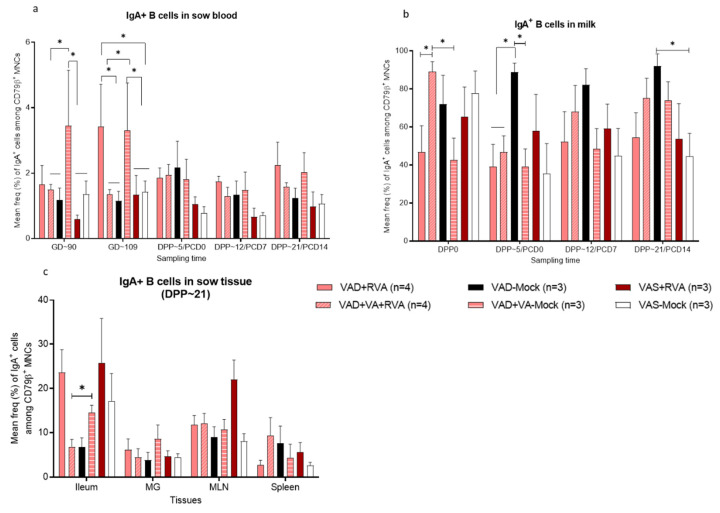
Mean frequencies (%) of IgA^+^ B cell subpopulations among B lymphocytes (CD79β^+^) in blood (**a**), milk (**b**) and tissues (Ileum, MNL, spleen and MG) (**c**) of RVA (OSU)/mock inoculated sows fed VAS and VAD(±VA) diets during gestation and postpartum. (MLN, mesenteric lymph nodes; MG, mammary gland) (* *p* < 0.05).

**Figure 9 viruses-14-02354-f009:**
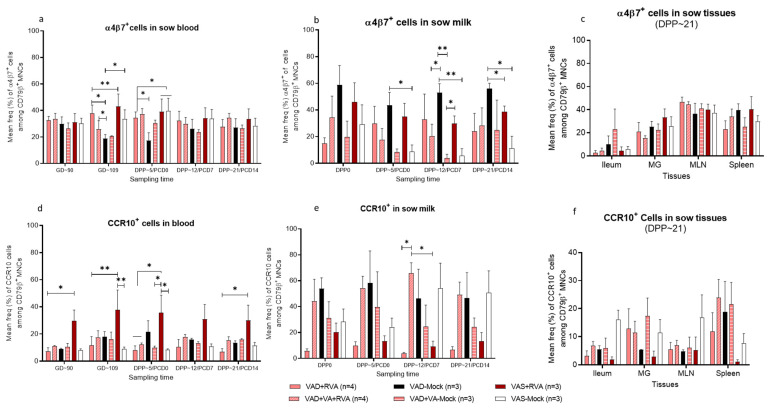
Mean frequencies (%) of α4β7^+^ (**Upper**) and CCR10^+^ (**Lower**) cell subpopulations among B lymphocytes (CD79β^+^) in blood (**a**,**d**), milk (**b**,**e**) and tissues (**c**,**f**) of RVA (OSU) infected sows fed VAS and VAD (±VA) diets during gestation and postpartum. (* *p* < 0.05; ** *p* < 0.01) (MLN, mesenteric lymph nodes; MG, mammary gland).

**Table 1 viruses-14-02354-t001:** Piglet diarrhea and RVA RNA peak shedding data.

Piglet Groups	No. of Piglets	Avg Peak RVA RNA Titer (Log 10 GE/mL)	% of Pigs with *^y^* Diarrhea (Score = ≥2)	Mean No. of Days to Onset of Diarrhea	*^z^* Mean Cumulative Fecal Score
VAD + RVA	52	7.05 *^ab^*	5.1	2	1.6 ^b^
VAD-Mock	26	15.02 *^a^*	36.8	2	4.8 ^a^
VAD + VA + RVA	52	4.14 *^b^*	7	2	1.4 ^b^
VAD + VA-Mock	27	9.58 *^ab^*	18.5	2	2.8 ^c^
VAS + RVA	20	4.06 *^b^*	0	2	1.1 ^b^
VAS-Mock	17	7.53 *^ab^*	23	2	2.5 ^c^

*y* Diarrhea is defined as fecal score ≥2. Fecal diarrhea was scored as follows: 0, normal; 1, pasty; 2, semiliquid; 3, liquid. *z* Mean cumulative fecal score [sum of fecal consistency score days post inoculation (PCD0-PCD12)/N], where N is the number of pigs receiving the inoculation. *a*, *b*, *c* represents significant differences among the groups within the column (*p* ≤ 0.05). *ab* represents no significant differences to any group within the column.

## Data Availability

Not applicable.

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
