# Peer review of "Rotavirus A Inoculation and Oral Vitamin A Supplementation of Vitamin A Deficient Pregnant Sows Enhances Maternal Adaptive Immunity and Passive Protection of Piglets against Virulent Rotavirus A"

_viruses, 2022, doi:10.3390/v14112354_

Round 1

Reviewer 1 Report

Authors have done interesting study on role of vitamin A in Rotavirus A infection. It is important to have the study like this where role of vitamins can be elucidate experimentally.

The parameters and presentation are nicely done.

The groups and challenge study were perfectly executed.

The cellular role is well identified.

Authors are suggested to do the study like this with combination of minerals or even with vitamin C and K in their future study, if possible.

Author Response

We thank you for taking time to read our manuscript for the overall positive evaluation. We are looking forward to incorporating your ideas to our future research.

Reviewer 2 Report

In this manuscript, the author aim to determind that Va has a positive effect on the immunization of RVA vaccine. Although the author demonstrated that Va enhanced maternal adaptive immunity and musocal immunity of rotavirus A inoculation sows, the biggest problem is that the sows were the RVA seropositive sows, which were able to influence the shedding and clinical signs. There are some concerns are listed below:

1. Whether the main challenge and the techniques for evaluation and quantification of passive maternal immunity proposed by author are resolved in this manuscript?

2. Line 96, however, influenza-specific IgG Ab response in serum was not affected by VAD. What is the IgG Ab response? It is the antigen-antibody response?

3. Line 257, there is no data to show that the weight gain of the piglets of mock VAD (±VA) sows was decreased throughout the experiment.

4. The RVA antibody data of RVA seropositive sows showed be provided.

Author Response

We thank you for taking time to read and critique our manuscript. Please see attached document for our reply. 

In this manuscript, the author aim to determind that Va has a positive effect on the immunization of RVA vaccine. Although the author demonstrated that Va enhanced maternal adaptive immunity and musocal immunity of rotavirus A inoculation sows, the biggest problem is that the sows were the RVA seropositive sows, which were able to influence the shedding and clinical signs. There are some concerns are listed below.

  1. Whether the main challenge and the techniques for evaluation and quantification of passive maternal immunity proposed by author are resolved in this manuscript?

Response 1: Our techniques, including sow RVA oral inoculation dose, VAD induction and piglet RVA challenge timing/dose, as well as the sampling strategy and time points are designed and validated to ensure that we can induce and detect statistically significant differences in clinical and immunological parameters in seropositive sows and their piglets. This strategy builds up on our previous ample expertise in studying mucosal and humoral immune responses to RVA in VAD and normal conventional and germfree pigs. 

While we concur with the reviewer’s statement that all sows used in our experiment are seropositive, we see it as one of the strengths of our research design as it mirrors the real-world scenario when 100% of adult swine (similar to adult humans) are RVA seropositive. Using seronegative sows (if they were available) would make our results hard to translate into field practice as we would be evaluating primary immune response instead of memory (anamnestic) as in this manuscript. Our goal was to demonstrate that we can boost sow immune responses improving their piglet passive protection.

  1. Line 96, however, influenza-specific IgG Ab response in serum was not affected by VAD. What is the IgG Ab response? It is the antigen-antibody response?

Response 2. IgA Ab response is one of the main Ab found in mucosal linings including the gut, respiratory tract, and adequate VA levels have been previously shown to be essential for optimal IgA Ab responses. In contrast, IgG Ab is mainly found in blood and protects against systemic viruses and bacteria and may or may not be affected by VAD.

  1. Line 257, there is no data to show that the weight gain of the piglets of mock VAD (±VA) sows was decreased throughout the experiment.

Response 3. We have edited to reflect that weight gain was decreased in piglets of VAD-mock sows (Figure 3a).

  1. The RVA antibody data of RVA seropositive sows showed be provided.

Response 4. The RVA specific- IgA and -IgG antibody titer data for all the sows are shown in Figure 5